# The *C. elegans* neural editome reveals an ADAR target mRNA required for proper chemotaxis

Sarah N Deffit[1], Brian A Yee[2], Aidan C Manning[1], Suba Rajendren[3], Pranathi Vadlamani[1], Emily C Wheeler[2], Alain Domissy[2], Michael C Washburn[3], Gene W Yeo[2,4,5]*, Heather A Hundley[1]*

[1]Medical Sciences Program, Indiana University, Bloomington, Indiana; [2]Department of Cellular and Molecular Medicine, Stem Cell Program and Institute for Genomic Medicine, University of California at San Diego, San Diego, United States; [3]Department of Biology, Indiana University, Bloomington, Indiana; [4]Molecular Engineering Laboratory, Agency for Science, Technology and Research, Singapore, Singapore; [5]Department of Physiology, Yong Loo Lin School of Medicine, National University of Singapore, Singapore, Singapore

*For correspondence:
geneyeo@ucsd.edu (GWY);
hahundle@indiana.edu (HAH)

**Competing interests:** The authors declare that no competing interests exist.

**Abstract** ADAR proteins alter gene expression both by catalyzing adenosine (A) to inosine (I) RNA editing and binding to regulatory elements in target RNAs. Loss of ADARs affects neuronal function in all animals studied to date. *Caenorhabditis elegans* lacking ADARs exhibit reduced chemotaxis, but the targets responsible for this phenotype remain unknown. To identify critical neural ADAR targets in *C. elegans*, we performed an unbiased assessment of the effects of ADR-2, the only A-to-I editing enzyme in *C. elegans*, on the neural transcriptome. Development and implementation of publicly available software, *SAILOR,* identified 7361 A-to-I editing events across the neural transcriptome. Intersecting the neural editome with *adr-2* associated gene expression changes, revealed an edited mRNA, *clec-41*, whose neural expression is dependent on deamination. Restoring *clec-41* expression in *adr-2* deficient neural cells rescued the chemotaxis defect, providing the first evidence that neuronal phenotypes of ADAR mutants can be caused by altered gene expression.
DOI: https://doi.org/10.7554/eLife.28625.001

## Introduction

The complexity of post-transcriptional gene regulation has exponentially expanded in recent years as the repertoire of RNA binding proteins (RBPs) in an organism as well as their binding sites have been uncovered (*Castello et al., 2012*; *Wessels et al., 2016*; *Beckmann et al., 2015*; *Brannan et al., 2016*). However, connecting this information to how different RBPs alter mRNA splicing, stability, as well as translation is only beginning to be elucidated, and likely varies in different cell- and tissue-types as well as during development. Adenosine deaminases that act on RNA (ADAR) are RBPs best known for converting adenosine (A) to inosine (I) within double-stranded RNA (dsRNA), including double-stranded regions of transcripts and small RNA precursors (*Nishikura, 2010*; *Goodman et al., 2012*). Transcriptome-wide identification of A-to-I editing sites from a number of species has indicated that RNA editing is prevalent in animal transcriptomes both in coding and non-coding regions of the genome, but the genomic distribution of these editing sites vary in different organisms (*Savva et al., 2016*). As inosine is a biological mimic of guanosine, A-to-I editing has the potential to affect gene expression by altering coding potential, splice site selection, and/or small RNA binding (*Deffit and Hundley, 2016*; *Nishikura, 2016*; *Tajaddod et al., 2016*). In

**eLife digest** DNA is the blueprint that tells each cell in an organism how it should operate. It encodes the instructions to make proteins and other molecules. To make a protein, a section of DNA known as a gene is used as a template to make molecules known as messenger ribonucleic acids (or mRNAs for short). The message in RNA consists of a series of individual letters, known as nucleotides, that tell the cell how much of a protein should be produced (referred to as gene expression) as well as the specific activities of each protein.

The letters in mRNAs can be changed in specific cells and at certain points in development through a process known as RNA editing. This process is essential for animals to grow and develop normally and for the brain to work properly. Errors in RNA editing are found in patients suffering from a variety of neuropathological diseases, including Alzheimer's disease, depression and brain tumors. Humans have millions of editing sites that are predicted to affect gene expression. However, many studies of RNA editing have only focused on the changes that alter protein activity.

The ADAR proteins carry out a specific type of RNA editing in animals. In a microscopic worm known as *Caenorhabditis elegans* the loss of an ADAR protein called ADR-2 reduces the ability of the worm to move in response to chemicals, a process known as chemotaxis. Deffit et al. found that loss of ADR-2 affected the expression of over 150 genes in the nervous system of the worm. To identify which letters in the mRNAs were edited in the nervous system, Deffit et al. developed a new publically available software program called SAILOR (software for accurately identifying locations of RNA editing). This program can be used to detect RNA editing in any cell, tissue or organism.

By combining the experimental and computational approaches, Deffit et al. were able to identify a gene that was edited in normal worms and expressed at lower levels in the mutant worms. Increasing the expression of just this one of gene in the mutant worms restored the worms' ability to move towards a chemical "scent".

Together, these findings suggest that when studying human neuropathological diseases we should consider the effect of RNA editing on the amount of gene expression as well as protein activity. Future work should investigate the importance of RNA editing in controlling gene expression in other diseases including cancers.

DOI: https://doi.org/10.7554/eLife.28625.002

addition, recent studies have shown mammalian ADARs play editing-independent roles through binding to mRNAs and influencing the association of other RBPs on the same target RNA (*Anantharaman et al., 2017*; *Wang et al., 2013*; *Bahn et al., 2015*).

One of the primary biological functions of ADARs is to promote proper neuronal function (*Li and Church, 2013*; *Behm and Öhman, 2016*). The mammalian brain contains the highest level of inosine, and A-to-I editing within coding regions of specific human transcripts, such as ion channels and receptors, alters the physiological properties of the encoded proteins, a requisite for proper neuronal function (*Paul and Bass, 1998*; *Tariq and Jantsch, 2012*; *Rosenthal and Seeburg, 2012*). Consistent with this important role of ADARs in the mammalian nervous system, alterations in ADAR protein levels and editing activity have been observed in human neuropathological diseases, including Alzheimer's disease, amyotrophic lateral sclerosis and many cancers, including brain tumors (*Mannion et al., 2015*; *Bajad et al., 2017*; *Tomaselli et al., 2014*).

Studies in model organisms provide additional evidence for a critical role for ADARs within the nervous system. Mice lacking *ADAR2* die of epileptic seizures by postnatal day 20 due to under-editing of one adenosine within the coding region of the glutamate receptor, which results in excessive calcium influx and neuronal excitocity (*Higuchi et al., 2000*). Loss of *Drosophila melanogaster* ADAR (*dADAR*) results in severe behavioral abnormalities, including extreme uncoordination, tremors, and a lack of courtship, as well as age-dependent neurodegeneration (*Palladino et al., 2000*). Similar to the mouse and fly model organisms, loss of ADARs in *C. elegans* results in abnormal neuronal function as evidenced by defective chemotaxis to a number of volatile chemicals sensed by the AWA and AWC neurons (*Tonkin et al., 2002*). Loss of *C. elegans* ADARs likely affects the ability of the neurons to sense these chemicals as the chemotaxis defect is less severe with increasing doses

of these volatiles (*Tonkin et al., 2002*). To date, the targets responsible for the chemosensory defects of *C. elegans* ADARs remain unknown.

In the present study, we sought to understand the neurobiological effects of RNA editing in worms by dissecting the neural gene regulatory role of ADR-2, the only A-to-I editing enzyme in *C. elegans*. Though multiple studies have used high-throughput sequencing to assess the editome and the role of ADR-2, these studies have been limited to analyzing RNA isolated from whole worms (*Whipple et al., 2015*; *Zhao et al., 2015*; *Goldstein et al., 2017*). However, a recently developed method utilized chemomechanical disruption of worms followed by fluorescent activated cell sorting (FACS) to obtain cells of interest from whole worms (*Spencer et al., 2014*; *Kaletsky et al., 2016*). By expressing a fluorescent marker in neural cells, we used this method to isolate and sequence the transcriptome of neural cells from wild-type and *adr-2* deficient worms. We performed the first unbiased tissue-specific assessment of RNA editing in *C. elegans*. High-throughput sequencing combined with detection of A-to-I editing events using our newly developed *SAILOR* software revealed over 7300 editing sites in the neural editome.

Additionally, a differential expression analysis identified 169 genes whose expression was changed upon loss of *adr-2*. To identify potential ADR-2 targets responsible for defects in chemotaxis, our study further focused on genes known to regulate this biological process. Here, *clec-41*, a gene previously found to be important for proper worm locomotion (*Simmer et al., 2003*), was found to be expressed and edited within the 3' untranslated region (UTR) in neural cells. In addition, *clec-41* mRNA transcripts were differentially expressed in neural cells lacking *adr-2*. Strikingly, transgenic expression of *clec-41* in neural cells of *adr-2* deficient worms was sufficient to rescue the aberrant chemotaxis of these animals. Furthermore, expression of a mutant ADR-2 protein incapable of deamination was not sufficient for proper *clec-41* expression or chemotaxis, indicating deamination is required for both gene regulation and chemotaxis. In sum, this is the first study to link noncoding A-to-I editing and altered expression of a specific transcript with a neurological consequence resulting from loss of ADARs.

## Results

### A-to-I editing in the neural system of worms primarily occurs in non-coding regions

To determine the role of ADR-2 in neural cells, the neural transcriptome of *C. elegans* was isolated from wild-type and *adr-2* deficient worms. To accomplish this, larvae at the first stage of development (L1 larvae) were subjected to chemomechanical disruption followed by FACS (*Spencer et al., 2014*). Both wild-type and *adr-2(-)* strains express green fluorescent protein (GFP) driven by the pan-neural promoter, *rab-3*, that allows for isolation of neural cells. This technique is robust in dissociating the larvae to single cells, while also resulting in mostly live cells (*Figure 1A*). The FACS profile indicated that 22% of live cells were GFP positive, consistent with the proportion of neural cells in L1 worms (222/558) and similar to previously published studies using this technique (*Figure 1B*) (*Spencer et al., 2014*). Consistent with neural enrichment in the GFP+ cells, qRT-PCR of a known neural gene, *syntaxin* (*unc-64*) (*Saifee et al., 1998*) indicated an enrichment in the isolated neural cells compared to non-neural cells (*Figure 1C*), whereas expression of the muscle gene, *myo-3* (*Ardizzi and Epstein, 1987*), was depleted in the isolated neural cells compared to the non-neural cells (*Figure 1C*).

To understand the role of ADR-2 in the neural transcriptome, RNA-sequencing (RNA-seq) of poly-A$^+$ selected RNA isolated from wild-type (N2) and *adr-2(-)* neural cells was performed. To identify and estimate A-to-I editing events from the neural transcriptome of *C. elegans*, we developed the *SAILOR* (**S**oftware for **A**ccurately **I**dentifying **L**ocations **O**f **R**NA editing) software, a publicly available improvement in both speed and accuracy of the algorithm described in *Washburn et al. (2014)*. Briefly, strand-specific RNA-seq reads were aligned to the *C. elegans* genome (ce11) to identify single nucleotide changes between RNA and DNA (*Figure 2A*). To eliminate possible genomic mutations that mimic A-to-I editing events, known single nucleotide polymorphisms (SNPs) and single nucleotide variants (SNVs) contained within Wormbase (WS254) annotations were eliminated. To measure the accuracy of editing events called, a confidence score was assigned using a beta distribution that considers both read depth and editing site percentage at each site. Only sites with a

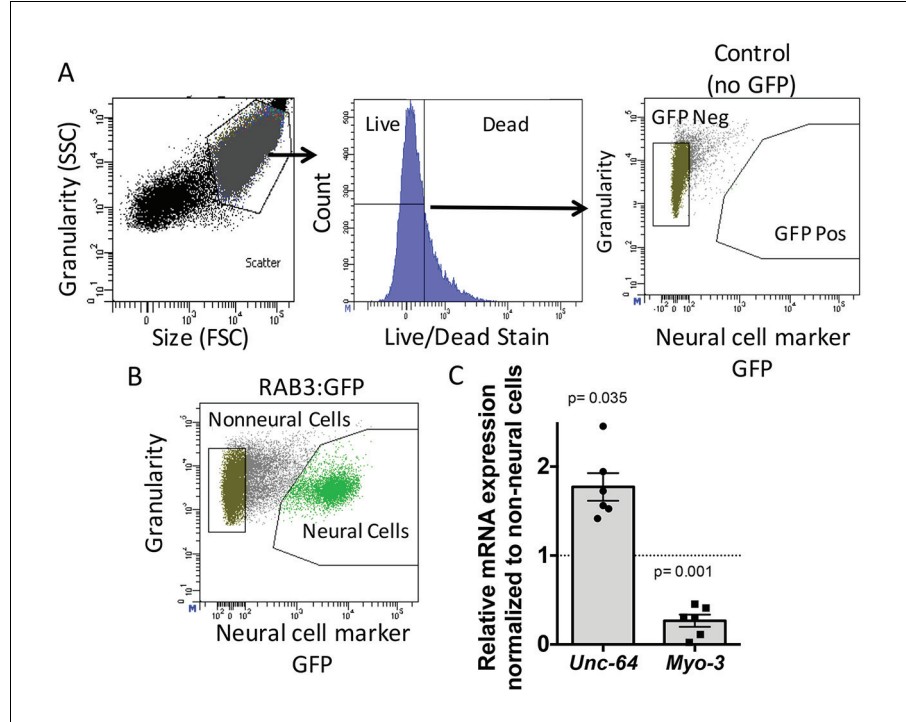

**Figure 1.** Neural cell isolation. (**A**) To establish gating parameters L1 worms were digested into a single cell suspension and stained with a live/dead stain. SSC (side scatter) and FSC (forward scatter) were used to select the single cell population and dead cells from this population were excluded. FACS was used to isolate single live neural (GFP) and non-neural cells from (**B**) transgenic *C. elegans* expressing GFP driven by the neural promoter (*rab3*). (**C**) mRNA expression of neural *syntaxin* (*unc-64*) and non-neural *myo-3* were determined by qRT-PCR relative to the housekeeping gene *gpd-3* for 6 independent biological replicates. Levels of mRNA expression in control non-neural cells were normalized and set to 1 (dotted line) and relative expression of neural mRNA was plotted with SEM. Student's t-test comparing non-neural to neural for each gene.
DOI: https://doi.org/10.7554/eLife.28625.003

confidence score greater than 99% were considered for downstream analyses. This approach identified only 28 editing events in the *adr-2(-)* neural transcriptome, whereas 7377 events were identified in the wild-type neural transcriptome. After subtracting the sites that were identified in the *adr-2(-)* neural transcriptome, 7361 A-to-I editing sites were predicted for the wild-type neural transcriptome (*Supplementary file 1*). The 7361 predicted editing sites mapped to 549 genes, 104 of which are novel edited targets (*Supplementary file 1*) (*Goldstein et al., 2017*).

To independently validate the predicted editing sites, Sanger sequencing assays were performed for 7 genes that had several predicted editing events either in introns or 3' untranslated regions (UTRs). As low level editing events are difficult to discern in Sanger sequencing assays, the validation focused on sites that were identified as having greater than 10% editing in the RNA-seq data. Importantly, 86 of 86 predicted editing sites (excluding 4 sites in which Sanger sequencing chromatogram had too much noise to accurately determine the presence of a true editing event) were confirmed (*Figure 2B* and *Supplementary file 1*). This data indicates the false discovery rate (FDR) of *SAILOR* is less than 1%, which is a significant improvement over the 6% FDR reported in our previously described algorithm (*Washburn et al., 2014*). Furthermore, Sanger sequencing identified 87 additional editing sites in these 7 genes, demonstrating *SAILOR* can identify A-to-I editing sites on highly edited transcripts (*Supplementary file 1*).

The initial assessment of the distribution of A-to-I editing events in the neural transcriptome indicated that nearly one-third of the 7361 editing sites were in intergenic regions. As the complete untranslated region (UTR) sequences are not annotated for all *C. elegans* genes and previous studies have indicated that many editing sites fall 1,000–2,000 base-pairs from the annotated UTRs (*Zhao et al., 2015*; *Whipple et al., 2015*), the genomic distribution of editing sites was re-analyzed

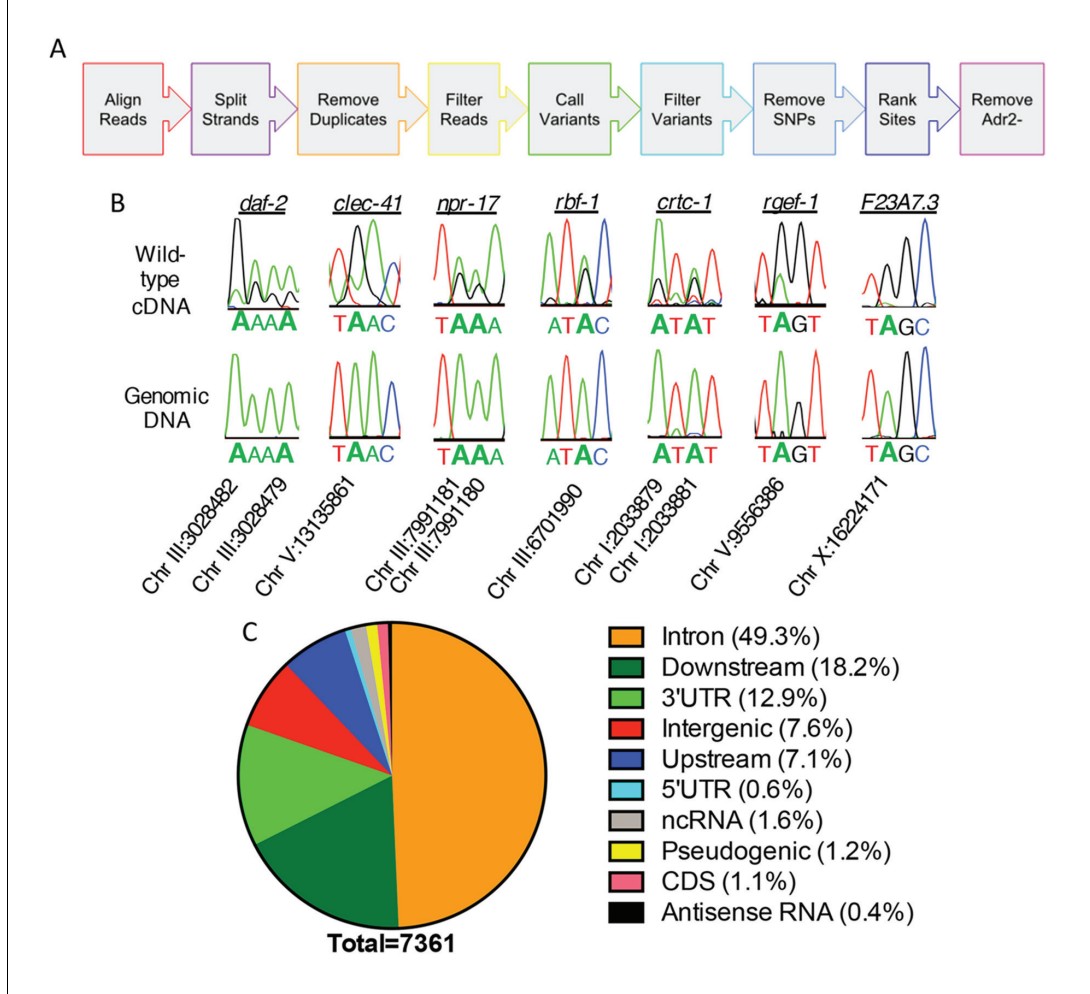

**Figure 2.** Neural editome of *C. elegans*. (a) RNA-seq libraries generated from *C. elegans* neural cells were sequenced using an Illumina NextSeq500 from 3 independent biological replicates. The data was processed with our *SAILOR* software with a series of steps (represented by each colored arrow) to identify A-to-I editing sites. (B) Sanger sequencing chromatograms of cDNA amplified from wild-type RNA and genomic DNA surrounding editing sites predicted by the bioinformatics pipeline. The specific gene analyzed is listed above the chromatograms and the chromosomal coordinates (ce11) for each editing site are listed below each chromatogram. The nucleotides at each position are represented with a different color (Green = Adenosine, Black = Guanosine, Blue = Cytidine, Red = Thymidine). A-to-I editing sites can be identified by peaks that are green (A) in the amplified genomic DNA and black (G) or a mixture of black (G) and green (A) in the cDNA. (C) Distribution of predicted neural A-to-I editing sites based on location in the genome (intron, coding sequence (CDS), 3' UTR, 5' UTR, noncoding RNA (ncRNA), antisense RNA, pseudogenic regions, 2000 bp upstream or downstream or intergenic).

DOI: https://doi.org/10.7554/eLife.28625.004

including annotations of 2000 base-pairs upstream and 2000 base-pairs downstream of a genic region (*Supplementary file 1*). This analysis indicated that 92.4% of the 7361 editing sites were within genic regions, with the vast majority of these events occurring in the noncoding regions of protein-coding genes, including introns and UTRs (*Figure 2C*). Importantly, analysis of the unique genes identified as editing targets revealed the same 549 genes using both genic assessments, consistent with the idea that the 2000 base-pair regions upstream and downstream of genic regions are likely unannotated 5' and 3' UTRs. It is interesting to note that in contrast to the identification of A-to-I editing events in coding regions of neuronally important mRNAs in squid, octopus, fly, and mammalian brains (*Alon et al., 2015*; *St Laurent et al., 2013*; *Ramaswami et al., 2013*; *Hwang et al., 2016*; *Graveley et al., 2011*; *Liscovitch-Brauer et al., 2017*), the worm neural RNA-seq data identified very few A-to-I editing events within coding regions. Of the over 7300 events identified, only 83 editing sites (approximately 1%) were in coding regions. These data are consistent

with the number of coding region editing sites previously identified from RNA isolated from whole-worms, and we identified nearly all of the same edited transcripts, including the heavily edited coding regions of histone genes (*Wu et al., 2011*). Together, this data indicates that re-coding editing events are not enriched in the *C. elegans* nervous system and demonstrates that the role of *C. elegans* ADARs in the nervous system is not to generate proteomic diversity, but rather to fine-tune gene expression.

## ADR-2 affects neural expression of target mRNAs independent of mRNA editing

To further assess the role of ADR-2 in regulating the neural transcriptome, the RNA-seq datasets from three independent biological replicates of wild-type and *adr-2(-)* neural cells were assessed for differential gene expression using DESeq (*Love et al., 2014*) (*Supplementary file 2*). This analysis identified 76 genes that were ≥2 fold upregulated in neural cells lacking *adr-2* as compared to wild-type neural cells and 93 genes that were ≥2 fold downregulated. Of these 169 differentially regulated genes, only four genes were identified as ADR-2 editing targets in neural cells, suggesting ADR-2 can also regulate expression of many genes via a mechanism that is independent of direct editing of the target mRNA. To validate the differential expression values obtained from the RNA-seq dataset, qRT-PCR was performed for two genes from each category (*Figure 3A and B*). Expression of these genes reflected the fold-change determined by the RNA-seq analysis. Interestingly, qRT-PCR analysis of the expression of these genes in RNA isolated from whole L1 larvae did not

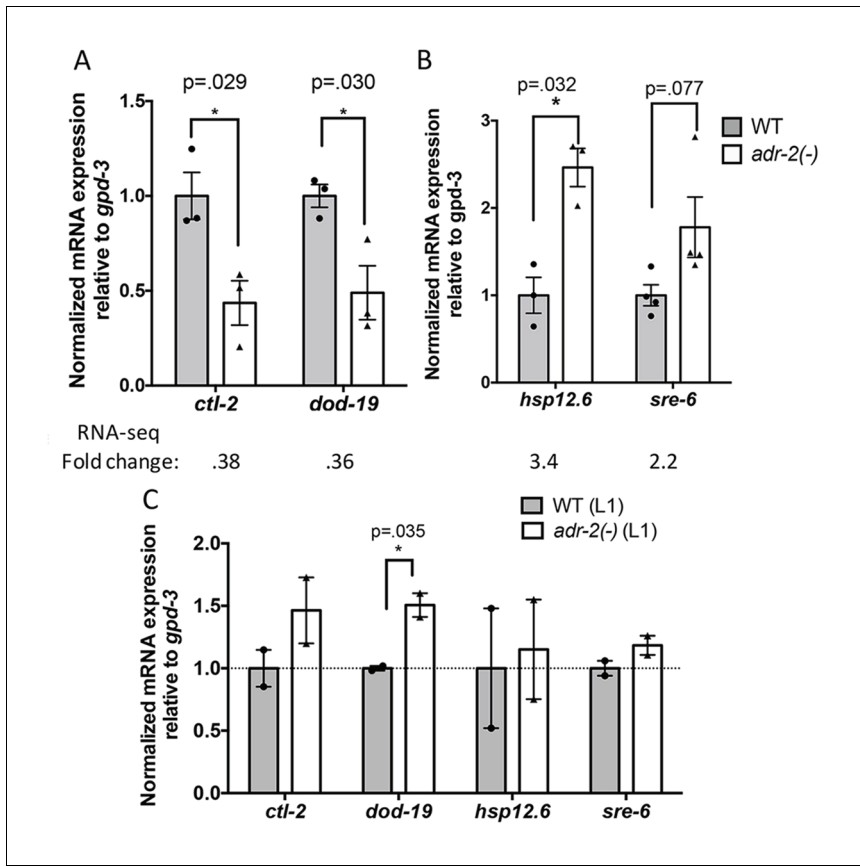

**Figure 3.** Differential gene expression in neural cells isolated from *adr-2(-)* worms. RNA was isolated (**A–B**) from wild-type (WT) and *adr-2(-)* neural cells (3 or 4 biological replicates) or (**C**) WT and *adr-2(-)* L1 whole worms (2 biological replicates). qRT-PCR was used to determine the relative expression of mRNA. Expression levels of the indicated genes were normalized to expression levels of the endogenous control *gpd-3* and plotted with SEM. Student's t-tests comparing WT to *adr-2(-)*, *p<0.05.
DOI: https://doi.org/10.7554/eLife.28625.005

follow the same trend (*Figure 3C*). This reveals that loss of *adr-2* regulates expression of genes at a tissue-specific level in *C. elegans* and suggests that these changes may not be observed in whole worm analysis.

## ADRs regulate editing and expression of *clec-41* within neural cells

Previous studies have shown that worms lacking *adr-2(-)* exhibit defects in chemotaxis (*Tonkin et al., 2002*), yet the underlying mechanism has yet to be determined. To attempt to elucidate ADR-2 target mRNAs that are important for proper neural function, the neural editome identified above was queried for genes involved in chemotaxis or locomotion. Editing sites were predicted in 82 genes that are associated with locomotion or chemotaxis GO terms (*Supplementary file 1*). To narrow down this list of potential neurobiologically important targets, we focused on previous phenotypic data which demonstrated that, similar to the loss of *adr-2*, loss of *adr-1* results in worms that are defective in sensing organic compounds (*Tonkin et al., 2002*). ADR-1 shares similar sequence and domain structure to ADR-2, though it lacks critical amino acids necessary for deamination (*Tonkin et al., 2002*). As ADR-1 promotes editing by ADR-2 at specific adenosines (*Washburn et al., 2014*), the similar phenotypes of *adr-1(-)* and *adr-2(-)* worms suggests that ADR-1 promotes editing by ADR-2 at sites important for proper chemotaxis. Using the assumption that ADR-1 regulates neurobiological transcripts of interest, the 82 edited genes involved in locomotion and chemotaxis were queried against a list of transcripts identified in a transcriptome-wide RNA immunoprecipitation of ADR-1 bound targets (unpublished data). Of the 82 edited genes that are involved in proper worm locomotion/chemotaxis, 28 were identified as bound by ADR-1 (*Supplementary file 1*). Of these genes, only one, *clec-41*, was also identified as differentially expressed in neural cells lacking *adr-2* (*Supplementary file 2*).

Focusing on *clec-41*, the role of ADR-1 in regulating *clec-41* expression and editing in neural cells was further explored. First, ADR-1 binding to *clec-41* was examined using a RNA immunoprecipitation assay for ADR-1 (*Figure 4A*). Analysis of two independent biological replicates indicated a >20 fold enrichment of *clec-41* mRNA in immunoprecipitations from wild-type worms compared to those lacking *adr-1*. Thus, consistent with the transcriptome-wide RIP-Seq, *clec-41* is a mRNA target of ADR-1. To test whether editing of *clec-41* was regulated by *adr-1*, RNA from *adr-1(-)* neural cells was isolated and compared to RNA from wild-type and *adr-2(-)* neural cells. RT-PCR and Sanger sequencing of the *clec-41* 3′ UTR was performed for all three strains. Consistent with *clec-41* transcripts being *bona fide* targets of adenosine deamination, the wild-type neural cells exhibited A-to-I editing at 40 individual adenosines within the *clec-41* 3′ UTR and this editing was absent in neural cells lacking *adr-2* (*Figure 4B* and *Figure 4—figure supplement 1A*). Compared to RNA from wild-type neural cells, editing of *clec-41* is significantly altered at 12 sites in RNA isolated from *adr-1(-)* neural cells (*Figure 4B* and *Figure 4—figure supplement 1A*). ADR-1 regulation of editing at these adenosines was not observed in RNA isolated from L1 worms (*Figure 4B* and *Figure 4—figure supplement 1B*). In sum, only 2 of the 12 ADR-1 regulated editing sites exhibited similar editing levels between the L1 and neural cell RNA, suggesting neural-specific regulation of editing at these sites by ADR-1.

To examine the role of *adr-1* in regulating *clec-41* expression, qRT-PCR of RNA from both isolated neural cells and L1 worms was performed. Loss of *adr-1* resulted in a three-fold reduction of *clec-41* expression in neural cells (*Figure 4C*). Consistent with the differential expression data, loss of *adr-2* resulted in a similar three-fold downregulation and the regulatory effect of either *adr-1* or *adr-2* cannot be observed in RNA isolated from L1 worms (*Figure 4C and D*). This suggests that both ADR-1 and ADR-2 are required for proper *clec-41* expression and editing in neural cells. This is consistent with previous data that indicated a similar chemotaxis defect for worms lacking either *adr-1* or *adr-2* and worms lacking both *adr-1* and *adr-2* (*Tonkin et al., 2002*). Despite the fact that the similarity in chemotaxis phenotypes has been known for over a decade, this study is the first to identify an editing target whose expression is regulated by both *C. elegans* ADARs. Furthermore, this study brings into light the need to assess editing and gene expression at a tissue-specific level in *C. elegans*, as this molecular phenotype cannot be observed in RNA isolated from L1 worms.

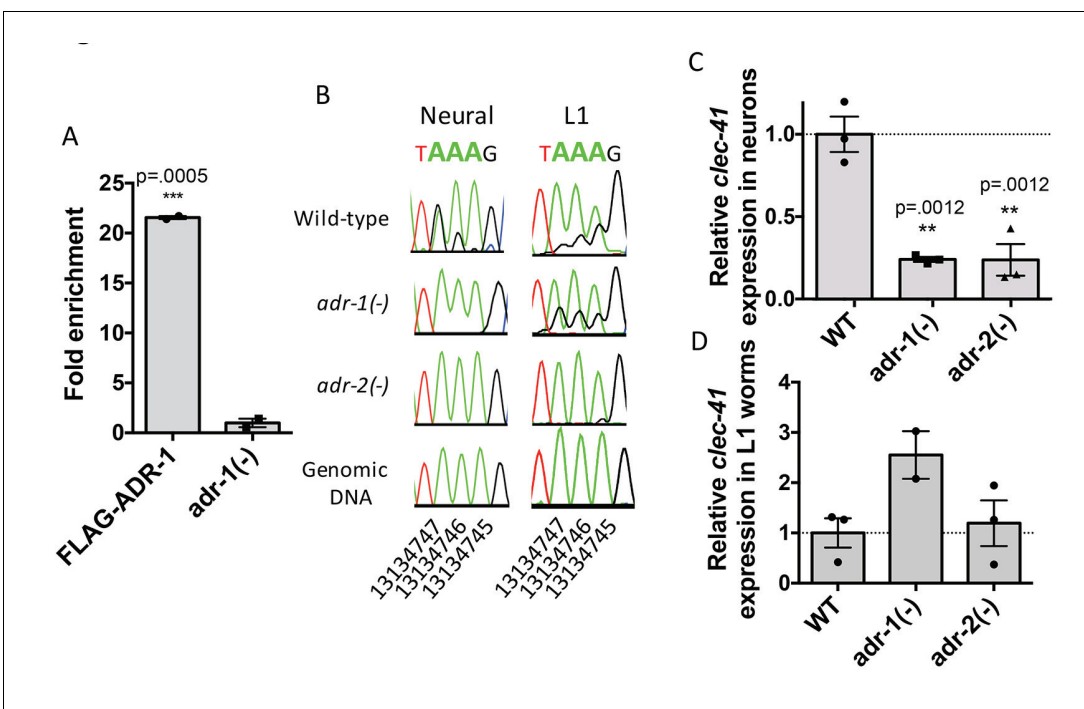

**Figure 4.** Neural A-to-I RNA editing and expression of *clec-41* is regulated by ADR-1 and ADR-2. (**A**) Lysates from *adr-1(-)* worms and worms expressing FLAG-ADR-1 were subjected to a FLAG immunoprecipitation (IP). qRT-PCR was performed on both RNA from the input lysates as well as the IP samples. The levels of *clec-41* in the IP samples was divided by the level of *clec-41* in the lysate and the fold enrichment of this ratio for FLAG-ADR-1 normalized to the negative control *adr-1(-)* was determined. The average of two independent biological replicates is plotted with error bars representing the SEM. Student's t-test, ***p<0.001. (**B**) Sanger sequencing chromatograms of *clec-41* genomic DNA or cDNA amplified from the indicated strains. The chromosomal coordinates (ce11) for the edited adenosines in the wild-type cDNA are indicated below the chromatograms, representative from three (Neural) or 2 (L1) independent biological replicates (Quantification of all editing sites can be seen in *Figure 4—figure supplement 1*) RNA isolated from neural cells (**C**) or L1 whole worms (**D**) for the indicated strains was subjected to reverse transcription and qRT-PCR to determine levels of *clec-41* from three independent biological replicates. The average expression of *clec-41* relative to the house-keeping gene, *gpd-3* were normalized to WT and plotted with SEM. One-way ANOVA followed by Dunnett's Multiple Comparisons Correction, **p<0.01.

DOI: https://doi.org/10.7554/eLife.28625.006

The following figure supplement is available for figure 4:

**Figure supplement 1.** Changes in editing of the *clec-41* 3' UTR upon loss of *adr-1*.

DOI: https://doi.org/10.7554/eLife.28625.007

## Transgenic *clec-41* expression within neural cells rescues the chemotaxis defect of *adr-2* deficient worms

As our gene regulatory data indicates that both *C. elegans* ADARs are important for proper *clec-41* expression and a previous genome-wide screen suggested *clec-41* was important for proper loco-motion/chemotaxis, we sought to directly assess the role of *clec-41* in regulating the chemotaxis defects associated with loss of ADARs. To test this, a worm line expressing *clec-41* under the control of the pan-neuronal *rab-3* promoter (*rab3p*) was generated. To visually follow the *clec-41* transgene, the worms were co-injected with a *rab3p::GFP* reporter. As the chemotactic effects of restoring *clec-41* expression to *adr-2(-)* worms were of interest, the transgenic wild-type worm generated by micro-injection was crossed to *C. elegans adr-1(tm668); adr-2(ok735)* mutant male worms. After identifying progeny that were wild-type for the *adr-1* locus, worms homozygous for the *adr-2(ok735)* mutation and carrying the *rab3p::clec-41* transgene as well as wild-type worms carrying the transgenes were isolated from the F2 progeny. Neural expression of *clec-41* from the transgene in both the wild-type and *adr-2(-)* worms was confirmed by performing qRT-PCR on neural cells isolated from the

transgenic worms. Compared to wild-type worms, both transgenic lines express higher levels of *clec-41* in neural cells (*Figure 5A*).

As previous chemotaxis studies indicated that *adr-2(-)* worms are defective in sensing volatile organic chemicals that are sensed by AWA and AWC neurons (*Tonkin et al., 2002*), we tested whether transgenic expression of *clec-41* in neural cells altered chemotaxis to benzaldehyde (sensed by AWC) or trimethylthiazole (sensed by both AWA and AWC) (*Bargmann et al., 1993*). Here, plates containing normal worm growth media were marked as shown in *Figure 5B*. Immediately prior to

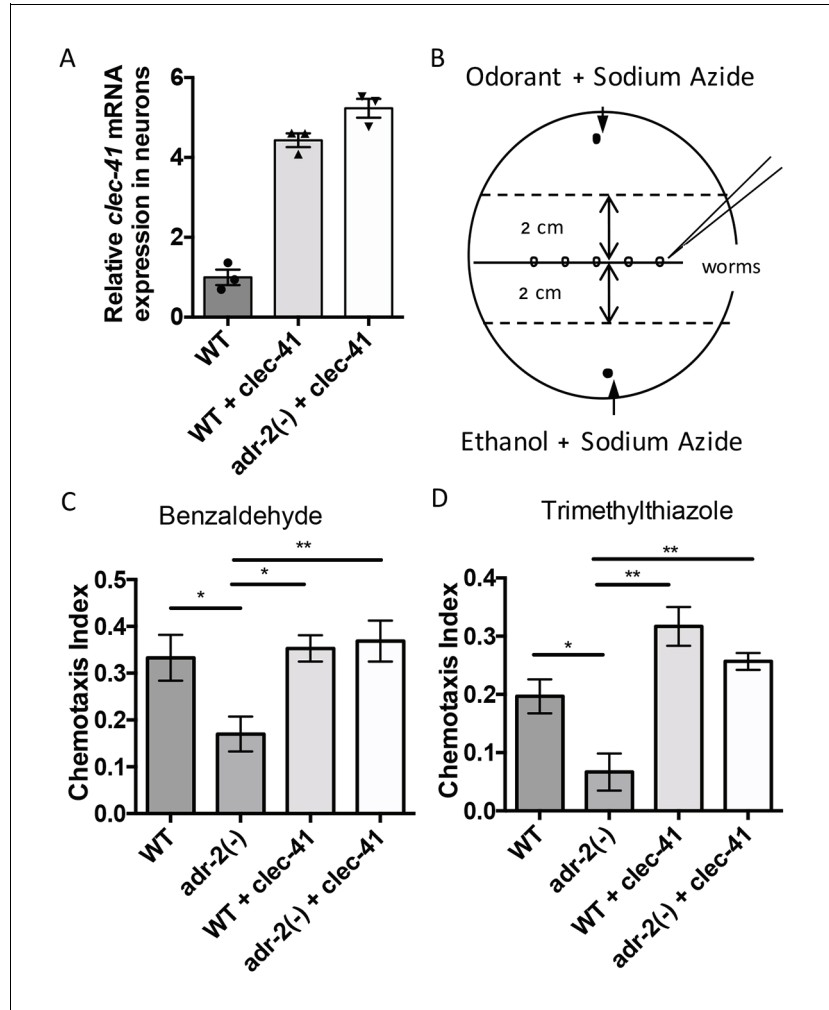

**Figure 5.** Rescue of *clec-41* expression in *adr-2(-)* neural cells prevents disruptions in chemotaxis. RNA was isolated from neural cells from wild-type (WT) as well as WT +*clec-41* and *adr-2(-)+clec-41* transgenic worms expressing *clec-41* using the pan-neural *rab-3* promoter for (**A**) qRT-PCR analysis of *clec-41* expression. The endogenous control *gpd-3* was used to normalize expression levels. (**B**) Chemotaxis assays used 60 cm plates. The chemoattractant (odorant) was spotted on one side and an ethanol control on the other. Worms were placed in the middle and allowed to migrate for 1 hr prior to counting (*Wang et al., 2016*) and the Chemotaxis Index of WT, *adr-2(-)* as well as WT +*clec-41* and *adr-2(-)+clec-41* to (**C**) Benzaldehdye (1:1000 dilution) or (**D**) Trimethylthiazole (1:10,000 dilution) was determined from 7 and 3 independent biological replicates, respectively. The chemotaxis index to trimethylthiazole (1:10,000 dilution) of worms expressing only the *clec-41 3' UTR* in neural cells or expressing *clec-41* in nonneural tissue was determined as a control (*Figure 5—figure supplement 1*). One-way ANOVA followed by Tukey's Multiple Comparisons Correction. *p<0.05, **p<0.01.

DOI: https://doi.org/10.7554/eLife.28625.008

The following figure supplement is available for figure 5:

**Figure supplement 1.** Neural expression of clec-41 gene required for proper chemotaxis.
DOI: https://doi.org/10.7554/eLife.28625.009

addition of worms, the paralysis agent sodium azide as well as a control (Ethanol) or the chemoattractant were spotted on opposite sides of the plate. Worms were placed in the center and allowed to move for one hour, after which the worms were counted to determine a chemotaxis index. Similar to previous results obtained using a different deletion allele of *adr-2*, worms lacking *adr-2(-) (adr-2 (ok735))* exhibited reduced chemotaxis indices to both benzaldehyde and trimethylthiazole compared to wild-type worms (***Figure 5C and D***). Strikingly, expression of *clec-41* within the neural cells of *adr-2(-)* worms led to a significant increase in the chemotaxis indices (***Figure 5C and D***). Importantly, the increased expression of *clec-41* within neural cells did not significantly increase the chemotaxis index over wild-type worms. In addition, expression of the *clec-41* 3' UTR in neural cells was not sufficient to restore chemotaxis to *adr-2(-)* worms nor was expression of *clec-41* in nonneural tissue (***Figure 5—figure supplement 1A and B***). This data suggests restoring CLEC-41 expression within neural cells is sufficient to prevent disruptions in chemotaxis specifically associated with loss of *adr-2*.

### A-to-I RNA editing is required for proper *clec-41* expression and chemotaxis

As ADAR family members are known to perform both editing-dependent and independent mechanisms of gene regulation, we sought to directly assess the role of A-to-I RNA editing in *clec-41* mRNA expression and chemotaxis. In this regards, we queried worm strains from The Million Mutation Project (***Thompson et al., 2013***) for missense mutations in conserved residues of the *adr-2* deaminase domain. The *adr-2(gk777511)* strain has a single nucleotide mutation that results in the incorporation of an arginine instead of a conserved glycine at amino acid 184 (G184R) and will hereafter be referred to as the *adr-2(G184R)* worms (***Figure 6A***). To determine whether the G184R mutation affects deaminase activity, RT-PCR and Sanger sequencing of the *clec-41* 3' UTR was performed on RNA isolated from wild-type, *adr-2(-)* and *adr-2(G184R)* worms. At all 40 adenosines edited in wild-type worms, editing was absent in *adr-2(G184R)* worms, similar to worms lacking *adr-2* (***Figure 6B***, data not shown). To examine whether the reduced editing of the ADR-2 G184R mutant was a result of loss of RNA binding, an ADR-2 RNA immunoprecipation assay was performed. Using a custom antibody that is specific for ADR-2 (***Figure 6—figure supplement 1***), similar levels of ADR-2 protein were immunoprecipitated from both wild-type and *adr-2(G184R)* worms, whereas no ADR-2 protein was immunoprecipitated from the *adr-2(-)* worms (***Figure 6C***). Analysis of two independent biological replicates indicated a >20 fold enrichment of *clec-41* mRNA in immunoprecipitations from both wild-type and *adr-2 (G184R)* worms compared to those lacking *adr-2,* indicating that the ADR-2 G184R mutant binds *clec-41* similar to wild-type ADR-2 (***Figure 6C***). To directly assess the impact of loss of RNA editing on *clec-41* neural gene expression, qRT-PCR of RNA from neural cells isolated from *adr-2 (G184R)* worms, as well as wild-type and *adr-2(-)* worms, was performed. Neural mRNA expression of *adr-2* was similar between the wild-type and *adr-2 (G184R)* neural cells (***Figure 6D***). However, *clec-41* expression was reduced three-fold in neural cells expressing the ADR-2 G184R mutant compared to wild-type neural cells, mirroring the *adr-2(-)* neural cells (***Figure 6E***) and suggesting deamination is required for proper neural expression of *clec-41*. Similarly, chemotaxis in the *adr-2 (G184R)* mutant worms was impaired compared to wildtype, further implicating A-to-I editing and *clec-41* expression in regulating this biological function (***Figure 6F***). This is the first study to link a phenotype associated with loss of *C. elegans* ADAR enzymes to editing-dependent regulation of a specific gene.

## Discussion

In this study, we have provided the first analysis of the tissue-specific role of ADR-2, both in A-to-I editing and in regulating neural gene expression in *C. elegans*. Combining neural cell isolation with RNA-sequencing and editing site detection with *SAILOR*, we identified over 7300 A-to-I editing sites in the *C. elegans* neural transcriptome, including 104 novel edited genes. In addition to identifying neural editing events, the neural RNA-seq data revealed 169 genes to be misregulated in worms lacking editing. The intersection of these two datasets lead to the identification of *clec-41*, a gene that was expressed three-fold lower in *adr-2(-)* neural cells and whose transcripts contained multiple 3' UTR editing events. Importantly, using a combination of transgenic worms and chemotaxis assays, we demonstrated that transgenic expression of *clec-41* in neural cells was sufficient to rescue the

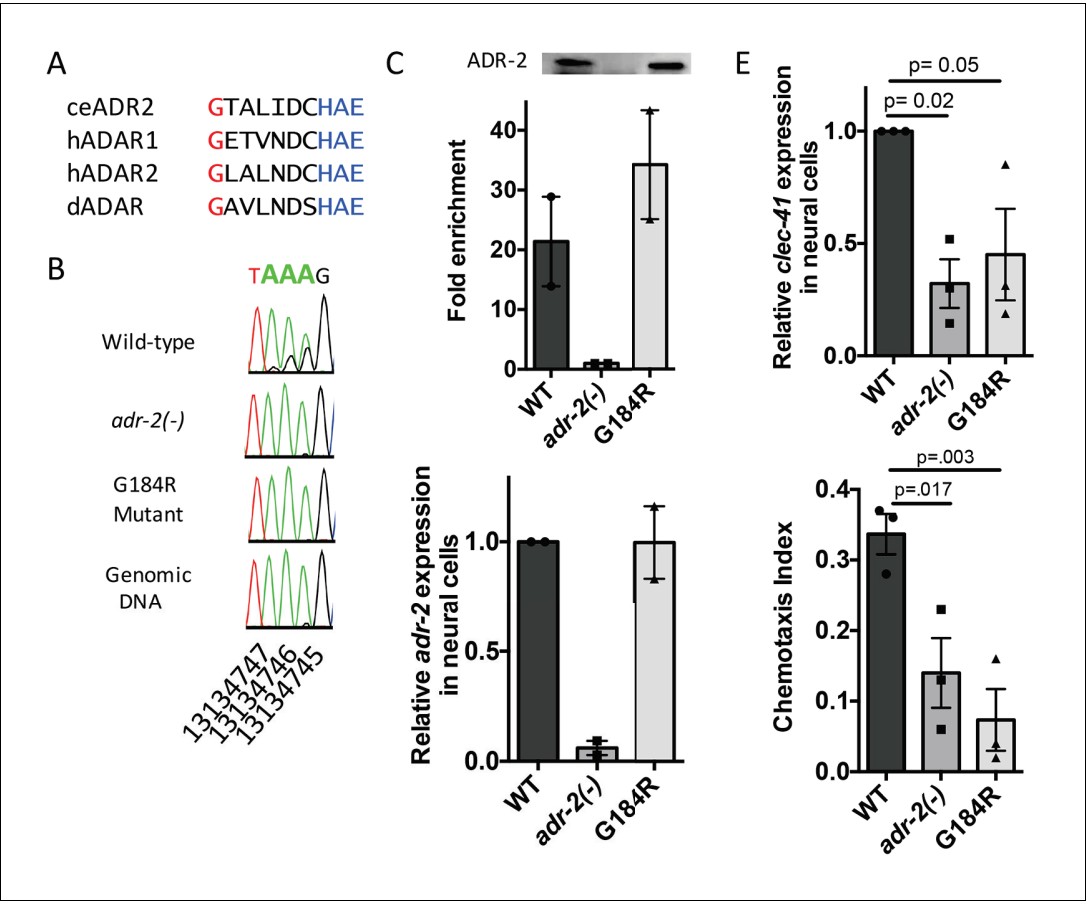

**Figure 6.** Deamination is required for both proper *clec-41* expression and proper chemotaxis. (**A**) Alignment of ADAR sequences from several species demonstrating conservation of the Glycine residue at position 184 in the *C. elegans* ADR-2 protein. This G is near the conserved HAE deamination motif and is mutated to arginine (**R**) in the *adr-2(G184R)* worms. (**B**) RNA from *adr-2(G184R)* worms was isolated and compared to RNA from wild-type and *adr-2(-)* worms. RT-PCR and Sanger sequencing of the *clec-41* 3' UTR was performed for all three strains. The chromosomal coordinates (ce11) for each editing site are listed below each chromatogram. The nucleotides at each position are represented with a different color (Green = Adenosine, Black = Guanosine, Blue = Cytidine, Red = Thymidine). A-to-I editing sites can be identified by peaks that are green (**A**) in the amplified genomic DNA and black (**G**) or a mixture of black (**G**) and green (**A**) in the cDNA. (**C**) Lysates from WT, *adr-2(-)* and *adr-2(G184R)* worms were subjected to an ADR-2 immunoprecipitation (IP). Western blotting of the IP samples with an ADR-2 antibody indicate similar levels of ADR-2 in the WT and *adr-2(G184R)* IPs. qRT-PCR was performed on both RNA from the input lysates as well as the IP samples. The levels of *clec-41* in the IP samples was divided by the level of *clec-41* in the lysate and the fold enrichment of this ratio for WT and *adr-2(G184R)* normalized to the negative control *adr-2(-)* was determined. (**D–E**) RNA was isolated from neural cells from wild-type (WT), *adr-2(-)* and *adr-2 (G184R)* worms. qRT-PCR analysis of (**D**) *adr-2* and (**E**) *clec-41* expression was analyzed in neural cells of all three strains. The endogenous control *gpd-3* was used to normalize expression levels. (**F**) Chemotaxis Index of WT, *adr-2(-)* as well as *adr-2(G184R)* worms to trimethylthiazole (1:10,000 dilution) was determined from 3 independent biological replicates. One-way ANOVA followed by Dunnett's Multiple Comparisons Correction.

DOI: https://doi.org/10.7554/eLife.28625.010

The following figure supplement is available for figure 6:

**Figure supplement 1.** Specificity of ADR-2 Antibody.

DOI: https://doi.org/10.7554/eLife.28625.011

aberrant chemotaxis of *adr-2* deficient worms and that editing is required for proper *clec-41* expression and chemotaxis.

Although ADARs play critical roles in editing codons of neuronal transcripts to generate the proteomic diversity needed for proper neuronal function in mammals, our data indicate the aberrant

behavioral phenotypes caused by lack of *C. elegans adr-2* are a result of altered neural gene expression rather than recoding events. Consistent with this idea, previous studies indicated that loss of critical components of the RNA interference (RNAi) pathway, namely *rde-1* or *rde-4*, can restore proper chemotaxis to *adr* mutant worms (*Tonkin and Bass, 2003*). However, previous attempts to identify genes antagonistically regulated by RNAi and ADARs were unable to pinpoint transcripts that underlie the chemotactic behavior (*Wu et al., 2011*). Interestingly, previous transcriptome-wide studies of gene expression in RNAi mutant worms identified elevated levels of *clec-41* in *dcr-1*, *rde-4* and *rde-1* mutant worms (*Welker et al., 2007*). Similarly, data from our lab has indicated that neural cells from *rde-4(ne299)* mutant worms have increased levels of *clec-41* compared to wildtype worms (unpublished data). Therefore, despite the conserved role of ADARs in regulating proper neurological function and essential roles for A-to-I editing within codons of neurologically important targets in certain organisms, our data indicate studies addressing the role of ADARs in the nervous system need to focus not only on direct editing targets but also on differentially expressed genes. In this regard, our analysis revealed 169 genes to be misregulated in *adr-2(-)* neural cells; however, A-to-I editing was only detected in the transcripts of four of these genes. At present, it is unclear how ADR-2 is regulating expression of these genes. Several recent studies have identified editing-independent mechanisms by which mammalian ADARs promote gene expression by either promoting or inhibiting binding of other RBPs. Specifically, human ADAR1 was shown to promote binding of the mRNA stabilizing RBP, HuR, while human ADAR2 was shown to inhibit the mRNA degrading RBP, poly(A) specific ribonuclease deadenylase (PARN) (*Anantharaman et al., 2017*; *Wang et al., 2013*). Future work will test whether the mRNAs with altered expression in *adr-2(-)* neural cells are directly bound by ADR-2 and if this binding alters other RBPs that influence the expression of these targets.

Our study identified altered expression of *clec-41* as a critical gene regulatory function of ADR-2. Strikingly, restoring *clec-41* expression in neural cells was sufficient to prevent disruptions in chemotaxis in *adr-2(-)* worms. It is currently unclear how CLEC-41, a predicted C-type lectin protein, functions to regulate proper chemotaxis. However, previous studies have determined that *clec-41* is regulated by the Protein Kinase D pathway, and that the DKF-2 B isoform of Protein Kinase D is expressed in ~20 chemosensory neurons, including the AWA and AWC neurons that are responsible for the aberrant chemotaxis defects observed in *adr-2* deficient worms (*Ren et al., 2009*; *Fu et al., 2009*). Future work aimed at determining whether *clec-41* expression is specifically altered in AWA and AWC neurons lacking *adr-2(-)* will be critical to understanding how CLEC-41 contributes to proper chemotaxis. In addition, experiments aimed at understanding the role of specific editing events and cellular factors, including the Protein Kinase D pathway, to regulate *clec-41* expression in neural cells will broaden our understanding of the gene regulatory functions of ADARs.

This study utilized novel methodology to assess A-to-I RNA editing in a specific tissue and developed a robust platform for easy identification of editing. The *SAILOR* software is publicly available and designed for ease of use to run with one single command, requiring only a BAM-formatted file of the sequence alignments, a FASTA-formatted reference genome sequence (of any organism or cell-type), and a BED-formatted file of known SNPs. Notably, *SAILOR* allows the user to specify a range of filtering criteria including: Non A-to-I mismatch rate, location of mismatches (to account for biases at the end of reads), and a minimum read coverage required to call variants. Users may relax any of these filtering criteria and/or pursue analysis of A-to-I editing sites with lower confidence scores. However, altering these criteria may result in a higher false discovery rate, therefore it is critical to utilize Sanger sequencing to verify a portion of all predicted sites. Our validation by Sanger sequencing revealed 87 additional editing sites not detected using our stringent SAILOR parameters, but either lowering read coverage requirements or decreasing confidence scores to 90% allowed for recovery of 54 of these sites. The complete environment is defined using Common Workflow Language (*Amstutz et al., 2016*) and packaged inside a Singularity (*Kurtzer et al., 2017*) container. This allows users the flexibility to run the entire workflow on a wide variety of platforms (workstation, cloud or HPC clusters) immediately after downloading the single executable file.

In sum, this study is the first of its kind in the RNA editing field to span from developing novel methodology for tissue-specific target identification to organismal behavior, significantly advancing our understanding of ADAR functions in neural cells. Our data indicates the necessity of assessing editing and gene expression at a tissue or cell-type specific level. Identification of the edited target *clec-41* as a critical regulator of chemotaxis in ADAR deficient worms would likely not have been possible without the neural-specific identification of gene expression changes as alterations in *clec-*

*41* editing and gene expression were not observed in RNA isolated from *adr* mutant whole worm extracts.

## Materials and methods

### Worm strains and maintenance

Worm strains were cultured on NGM plates seeded with *E. coli* OP50. The worm strains that express GFP driven by the pan-neural promoter, *rab-3*: BB76 (wild-type), BB77 (*adr-1(tm668)*), and BB78 (*adr-2(ok735)*) were previously published (*Hundley et al., 2008*). The worms used for the RNA immunoprecipitation were BB19 (*adr-1(tm668)*) and BB21 (*adr-1(tm668)* +blmEx1[3XFLAG-*adr-1* genomic, *rab3::gfp::unc-54*]), both previously published (*Washburn et al., 2014*). The VC40720 strain containing a deaminase mutant of ADR-2 (*gk777511*, chrIII:7232280 C > T) was obtained from the Caenorhabditis Genetics Center (CGC), but was originally created by the *C. elegans* Reverse Genetics Core Facility at the University of British Columbia. This strain was backcrossed eight times to generate strain HAH4 (*adr-2 (gk777511)*). This strain was then crossed to BB78 (*adr-2(ok735)*) heterozygous males and genotyping was used to identify progeny that were wild-type for the *adr-2* locus (named HAH8) or containing either the *adr-2(ok735)* mutation (named HAH9) or the *adr-2 (gk777511)* mutation (named HAH10).

### Transgenic worm line generation

Transgenic worm lines were generated by microinjection into the gonads of young adult N2 worms. The injection mix used for generating transgenic worms contained the following: 10 ng/μl of the transgene of interest, 20 ng/μl of the dominant marker, and 70 ng/μl of 1 kb DNA ladder (NEB, Ipswich, MA). Transgenic strains were maintained by passaging only worms with the dominant marker. The dominant marker used in this study was *rab3::gfp::unc-54*, which was previously described (*Hundley et al., 2008*). The transgene expressing the *clec-41* genomic locus was generated by amplifying *clec-41* from the start codon to 2000 bp downstream of the stop codon and inserting into a modified pBluescript SK plasmid containing the *rab3* promoter (~1200 bp) or the *myo-2* promoter (~1000 bp). The transgene expressing the *clec-41* 3'UTR downstream of RFP was generated by inserting RFP into a modified pBluescript SK plasmid and the *clec-41* 3' UTR downstream of RFP. Transgenic strains expressing these constructs were isolated using standard techniques following microinjection of the preceding plasmids into the gonads of adult hermaphrodites using the Bristol strain N2 and named blm9[*rab3::clec-41, rab3::gfp:unc-54 3' UTR*]. The injected *rab-3* promoter driven *clec-41* transgenic strain was crossed with BB21: *adr-1(tm668); adr-2(ok735)* and genotyping was used to identify progeny that were wild-type for the *adr-1* locus and contained either the wild-type *adr-2* locus (named HAH1: blm9[*rab3::clec-41, rab3::gfp:unc-54 3' UTR*]) or the mutant *adr-2 (ok735)* locus (named HAH2: *adr-2(ok735)* +blm9[*rab3::clec-41, rab3::gfp:unc-54 3' UTR*]). The injected *myo-2* driven *clec-41* transgenic and *rab3p::RFP::clec-41 3'UTR* transgenic strains were crossed with BB20 (*adr-2(ok735)*) and genotyping was used to identify progeny that contained either the wild-type *adr-2* locus (named HAH15 blm12[*myo-2::clec-41, rab3::gfp:unc-54 3' UTR*], HAH11 blm9[*rab3::rfp:clec-41 3' UTR, rab3::gfp:unc-54 3' UTR*] or the mutant *adr-2(ok735)* locus (named HAH16 *adr-2(ok735)* +blm12[*myo-2::clec-41, rab3::gfp:unc-54 3' UTR*], HAH12 *adr-2(ok735)* +blm9 [*rab3::rfp:clec-41 3' UTR, rab3::gfp:unc-54 3' UTR*].

### Isolation of neural cells

Neural cells were isolated from first larval stage (L1) worms using a previously published method (*Spencer et al., 2014*). Eggs were released from gravid adult worms by incubating in 0.5 M NaOH in 1.2% NaClO for 6 min. Eggs were thoroughly washed with M9 buffer (3 g $KH_2PO_4$, 6 g $Na_2HPO_4$, 5 g NaCl, 1 ml 1 M $MgSO_4$, $H_2O$ to 1 L) and cultured overnight at 20°C to synchronize worms in the first larval stage (L1). L1 worms were washed with $dH_2O$ and digested with freshly thawed SDS-DTT (200 mM DTT, 0.25% SDS, 20 mM HEPES, pH 8.0 3% sucrose, stored at −20°C) for 2 min on a nutator at room temperature. The digest was thoroughly washed with egg buffer (118 mM NaCl, 48 mM KCl, 2 mM $CaCl_2$, 2 mM $MgCl_2$ 25 mM HEPES, pH 7.3 adjusted osmolarity to 340 mOsm with sucrose) and further digested in freshly made 15 mg/ml Pronase E (Sigma-Aldrich, St. Louis, MO) dissolved in egg buffer. During the 20 min Pronase E digestion, the worms were mechanically disrupted

using a 200 µl pipette tip. The released cells were washed and resuspended in egg buffer. Cells were stained with the Near IR Live/Dead Fixable dye (Invitrogen, Carlsbad, CA) for 30 min prior to FACS sorting to isolate GFP expressing neural cells. Cells were sorted on a BD FACSAria II sorter and analyzed using FACSDiva 6.1.1 software (Indiana University Bloomington-Flow Cytometry Core Facility). Events were thresholded on FSC-A and SSC-A parameters at a threshold of 600 and 700 respectively. Cells were first gated on FSC-A vs. SSC-A, both analyzed on a log scale, and then doublet discrimination was done using SSC-H vs. SSC-W and FSC-H vs. FSC-W. Single cells were subsequently gated for Near IR Live/Dead Fixable dye negative populations, with final gating set on a GFP-A vs. SSC-A plot with gates for GFP positive and GFP negative cells. 200,000 cells were sorted into 15 ml conical tubes containing 1 ml Trizol. Both the sample and collected cells were kept at 4°C during and after the sort.

## RNA isolation

Total RNA was isolated from neural cells or from whole worms at the L1 stage using Trizol (Invitrogen) followed by treatment with DNase (Fisher Scientific, Hampton, NH) and purification using the RNeasy Extraction Kit (Qiagen, Hilden, Germany). qRT-PCR cDNA was synthesized from total RNA using Superscript III (Invitrogen) with both random hexomers (Fisher Scientific) and oligo dT (Fisher Scientific) primers. SybrFast Master Mix (KAPA, Wilmington, MA) as well as gene specific primers (Table 1) were used to quantitate gene expression on the Thermofisher Quantstudio 3.

## RNA library generation and sequencing

RNA libraries were created from RNA isolated from three independent biological replicates of FACS sorted neural cells from the wild-type (BB76) and adr-2(-) (BB78) worm strains. PolyA +beads (Invitrogen) were used to select for mRNA and libraries were generated using the KAPA Strand-Specific RNA Library Kit according to the manufacturer's instructions. The libraries were sequenced for SE150 cycles on an Illumina NextSeq500 in High Output mode (Indiana University Center for Genomics and Bioinformatics).

## RNA-Seq alignment

To accurately call editing sites, we improved upon our previously published bioinformatics algorithm as described in (Washburn et al., 2014), updated to accommodate for improved sequencing technology. Briefly, 150 bp single-end, stranded RNA-seq reads were trimmed of sequence adapters, polyA tails, and repetitive elements using cutadapt (v1.9.1), and aligned with STAR (v2.4.0i) against RepBase (v18) to remove repetitive elements. Reads were then aligned to ce11 using the following STAR parameters: [outFilterMultimapNmax 10, outFilterScoreMinOverLread: 0.66, outFiterMatchNminOverLread: 0.66, outFilterMismatchNmax: 10, outFilterMismatchNoverLmax: 0.3]. Sorting and duplicate removal were performed using Samtools 1.3.1. Triplicates for each condition (wild-type and adr2(-)) were merged into a single bam file prior to site calling to increase coverage to 98,161,600 reads for wild-type and 94,311,728 reads for adr-2(-).

## *SAILOR* software

*SAILOR* is publicly available on our github repository with documentation (https://github.com/yeo-lab/sailor; copy archived at https://github.com/elifesciences-publications/sailor) (Yee et al., 2017). The following parameters were used to further filter mapped reads prior to site calling: junction overhangs must be 10nt or longer, no insertions or deletions, no mutations within 5nt of the 3' end, not more than 1 non A-G (T-C in antisense) mutation. Samtools (v1.3.1) mpileup [-d 1000 -E -I -p -o -f -t DP,DV,DPR,INFO/DPR,DP4,SP] and bcftools (v1.2.1) call [-O -c -A -i -v] were used to pileup and call variants. Variants were further filtered for read coverage (minimum 5 reads after BAQ filtering from bcftools) and ensured that variants are all A-G (T-C in antisense). Variants sharing positions of SNPs contained within Wormbase (WS254) annotations were removed. Candidate editing sites were given a confidence score using a previously described Bayesian model (Bahn et al., 2012; Li et al., 2008) based on the number of supporting reads and the percent edited reads. Sites with less than 1% of reads edited and a 99% confidence score were not considered. Sites that saw 100% editing were flagged as possible SNPs. Editing sites called in the adr-2(-) datasets were removed from the final list of called sites. Annotations were gathered from Wormbase WS254. If two regions

**Table 1.** Sequences of all primers used in this study.

|  |  |  | Sequence |
|---|---|---|---|
| **qRT-PCR** | | | |
| unc-64 | | Forward | Gccattgatcacgacgagcaaggagccgga |
| | | Reverse | Ccagcaatatcgagttgtctctgaattcgtc |
| myo-3 | | Forward | ccagaagaatatcagacgctacttggac |
| | | Reverse | taacaataagctcttcttgctcctgtttg |
| gpd-3 | | Forward | ggaggagccaagaaggtc |
| | | Reverse | aagtggagcaaggcagtt |
| ctl-2 | | Forward | caagccaactcaaggagtgaagaatctcac |
| | | Reverse | catcttccatactggaaagtctcccttctc |
| dod-19 | | Forward | ccaggatatacgagcatcgattcgacaacc |
| | | Reverse | gaagctccaggatatctagtatctctcttg |
| hsp12.6 | | Forward | caatgtcctcgacgacgatgatcacttc |
| | | Reverse | gaatccttctttacattgtgctccatatgg |
| sre-6 | | Forward | gaaagatgctttgcgacatgtttcgctgg |
| | | Reverse | cgggcatcatgatagaaatcaagagaag |
| clec-41 | | Forward | actctggaagattctattccccaagc |
| | | Reverse | cgactgtaaatggaaattgatgcctgac |
| **Editing Assays** | | | |
| daf-2 | | RT primer | ctatttcgagcattgaggccgaattgaggc |
| | | Forward #1 | cgagaatgaatgaatattgtcagatgtcggag |
| | | Reverse #1 | cgagcgctacgtcgaattccaataactc |
| | | Forward #2 | gaaaatttggaagaaggtgagctggggg |
| | | Reverse #2 | ggtgggttaccgaaatttgagactttgc |
| clec-41 | | RT primer | acaccacgaaaaataattacagtgctggcc |
| | | Forward #1 | ctcaacagattcatctggccaaggttcagg |
| | | Reverse #1 | acaccacgaaaaataattacagtgctggcc |
| | | Forward #2 | ggttcaggattcagtgcaaatttttgggcg |
| | | Reverse #2 | agctcgagattactctacacttctcttctt |
| npr-17 | | RT primer | gctattgagttcattgagccatttacctggg |
| | | Forward #1 | ccaacttcaacaaagatatcgatcaaatcg |
| | | Reverse #1 | cattgagccatttacctgggaaaatgtggc |
| | | Forward #2 | gacgacaacaacaacagcttcaacagc |
| | | Reverse #2 | gttccgtataagtgtttacccagaagcg |
| rbf-1 | | RT primer | gtgtcaatgtgattgagccaaggctacctg |
| | | Forward #1 | ggggttattcaagtagtttcgcaac |
| | | Reverse #1 | tgagccaaggctacctgaatattttg |
| | | Forward #2 | ggggttattcaagtagtttcgcaac |
| | | Reverse #2 | gtgagaagaagaggaagatggaatattgatg |
| crtc-1 | | RT primer | ctctaatgccttcagattggcgccacctac |
| | | Forward #1 | ccaccaaacacccaacaactcattccatg |
| | | Reverse #1 | ccttcagattggcgccacctacaacatgg |
| rgef-1 | | RT primer | gaggaaagtgtgtggaagactggtg |
| | | Forward #1 | ggaagtacaccagatgaagaaattggtcttg |
| | | Reverse #1 | gcgtagagatcaaacaagtgggatagg |

*Table 1 continued on next page*

*Table 1 continued*

| | | Sequence |
|---|---|---|
| F23A7.3 | RT primer | ctaactgccaacaaacgactatctcaaatg |
| | Forward #1 | cacaactctcttgctggataggtccgaacg |
| | Reverse #1 | ctaactgccaacaaacgactatctcaaatg |
| | Forward #2 | gctggataggtccgaacgtcgtctaatg |
| | Reverse #2 | ctattctcatggagcatctgccattcc |

DOI: https://doi.org/10.7554/eLife.28625.012

overlapped, priority was assigned in the following order: [3'UTR, 5'UTR, CDS, Intron, mRNA, piRNA, ncRNA, tRNA, nc_primary_transcript, miRNA, snoRNA, pre_miRNA, lincRNA, scRNA, antisense_RNA, rRNA, miRNA_primary_transcript, scRNA, 2000 base-pairs downstream from gene, 2000 base-pairs upstream from gene, 2000 base-pairs downstream from ncRNA, 2000 base-pairs upstream from ncRNA, pseudogene, intergenic].

## Differential gene expression

150 bp single-end, stranded reads trimmed of sequence adapters, polyA tails, and repetitive elements using cutadapt (v1.9.1) and STAR (v2.4.0i) against RepBase (v18). Reads were then aligned to ce11 using the following parameters: [outFilterMultimapNmax 10, outFilterScoreMinOverLread: 0.66, outFiterMatchNminOverLread: 0.66, outFilterMismatchNmax: 10, outFilterMismatchNoverLmax: 0.3]. Mapped reads were sorted using Samtools 1.3.1. FeatureCounts was used to count mapped reads to Wormbase (WS254) gene annotations. WT and *adr-2(-)* triplicates were then normalized and compared for differential expression using DESeq2. Genes with a greater than 2-fold change in expression were reported.

## Editing assay

Editing sites were verified using total RNA isolated from neural cells or L1 stage worms. RNA was reverse transcribed using Thermoscript (Invitrogen), and PFX Platinum DNA Polymerase (Invitrogen) was used with gene specific primers for PCR amplification (*Table 1*). Negative controls without Thermoscript were conducted for each sample to ensure all DNA subjected to sequencing resulted from cDNA amplification. PCR products were gel purified and subjected to Sanger sequencing.

## GO analysis

To get the list of genes involved in chemotaxis, annotations derived from http://www.geneontology.org/ were intersected with the list of edited genes and filtered for the following GO terms: (GO:0040011 - locomotion, GO:0043058 - regulation of backward locomotion, GO:0040012 - regulation of locomotion, GO:0006935 - chemotaxis, GO:0040017 - positive regulation of locomotion, GO:0043059 - regulation of forward locomotion, GO:0050919 - negative chemotaxis).

## RNA Immunoprecipitation

The ADR-1 RNA immunoprecipitation was performed as previously described (*Washburn et al., 2014*). Briefly, after washing with IP buffer (50 mM HEPES [pH 7.4]; 70 mM K-Acetate, 5 mM Mg-Acetate, 0.05% NP-40, and 10% glycerol), worms were subjected to 3 J/cm$^2$ of UV radiation using the Spectrolinker (Spectronics, Westbury, NY) and stored at $-80°$C. To obtain cell lysates, frozen worms were ground with a mortar and pestle on dry ice. After thawing, the lysate was centrifuged and protein concentration was measured with Bradford reagent (Sigma-Aldrich). Five milligrams of extract was added to anti-FLAG magnetic beads (Sigma-Aldrich) that were washed with wash buffer (WB: 0.5 M NaCl, 160 mM Tris-HCl [pH 7.5]). After incubation for 1 hr at 4°C, the beads were washed with ice-cold WB, resuspended in low-salt WB (0.11 M NaCl), 1 µl RNasin (Promega, Madison, WI), and 0.5 µl of 20 mg/ml proteinase K (Sigma-Aldrich) and incubated at 42°C for 15 min to degrade protein and release bound RNA. Protein samples were subjected to SDS-PAGE and western blotting with a FLAG antibody (Sigma-Aldrich). RNA samples were isolated and qRT-PCR was performed as described above.

The ADR-2 RNA immunoprecipitation was performed in a similar manner as described for the ADR-1 RNA immunoprecipitation, except that a custom antibody to ADR-2 was incubated anti-Rabbit IgG magnetic Dynabeads (Fisher) before incubating with cell lysates. The ADR-2 antibody was produced by Cocalico Biologicals using partially purified ADR-2, which was tagged at the N-terminus with maltose binding protein (MBP) and purified from Hi5 insect cells using an amylose binding column. The amylose-bound MBP-ADR-2 was injected into rabbits for antibody generation.

## Chemotaxis assay

Adult worms were used to assess chemotaxis behavior using a method previously described (*Kowalski et al., 2014*). 10 µl of 1 M sodium azide was placed at both ends of the plate (see *Figure 5B* for location) and allowed to dry. Once dry, 10 µl of ethanol (control) or the chemoattractant diluted in ethanol were placed at either end and allowed to dry. The adult worms were washed 3x in M9 buffer. After the final wash, worms were resuspended in ~200 µl of M9 and 20 µl (50–200 worms per plate) was pipetted onto the center line of the plate. Chemotaxis to benzaldehyde (1:1000 dilution in ethanol) and trimethylthiazole (1:10,000 dilution) after 1 hr at room temperature was assessed and the chemotaxis index was determined using the formula below. Three technical replicate plates for each worm strain were used in each of the biological replicates.

$$\text{Chemotaxis index} = \frac{\text{Number of animals at attractant} - \text{Number of animals at control}}{\text{Total worms on plate}}$$

## Data deposit

Raw fastq files and their corresponding processed BAM alignments were made available and uploaded to the GEO database. Files generated from wild-type neural cells were labeled as: WT_1, WT_3, and WT_6 corresponding to the condition and original sample number. Files generated from *adr-2(-)* neural cells were labeled as: Adr2-_2, Adr2-_5, and Adr2-_7. A tabbed-separated file containing differential expression between WT and *adr-2(-)* is also available as diffexp.tsv. Editing sites, approximate fractions, confidence and region annotations made from the editing pipeline are available on GEO accession number GSE98869 as editing_calls.tsv. GEO link for reviewers to access raw data: https://www.ncbi.nlm.nih.gov/geo/query/acc.cgi?token=khqdcwemjhqbvgp&acc=GSE98869

## Acknowledgements

We thank Christiane Hassel from the IUB-Flow Cytometry Core for her assistance in sorting the neural cells and James Ford from the IUB-Center for Genomics and Bioinformatics for his assistance with the next generation sequencing. We also thank Boyko Kakaradov for laying the groundwork for the *SAILOR* method. This work was supported by funds to HAH from the American Cancer Society (RSG-15–051-RMC) and start-up funds from Indiana University School of Medicine as well as funds to SND from the National Institutes of Health (F32GM119257-01A1) and an Indiana Clinical Translational Science Institute (CTSI) Core Grant. This work was partially supported by grants from the NIH (HG004659 and NS075449) to GWY. ECW is supported by grants from the University of California, San Diego, Genetics Training Program (T32, GM008666) and the NSF Graduate Research Fellowship Program. Some strains were provided by the CGC, which is funded by NIH Office of Research Infrastructure Programs (P40 OD010440).

## Additional information

### Funding

| Funder | Grant reference number | Author |
|---|---|---|
| American Cancer Society | RSG-15-051-RMC | Heather A Hundley |
| Indiana Clinical and Translational Sciences Institute | | Sarah N Deffit |
| National Science Foundation | | Emily C Wheeler |
| National Institutes of Health | 1F32GM119257-01A1 | Sarah N Deffit |

| National Institutes of Health | T32GM00866 | Emily C Wheeler |
| National Institutes of Health | HG004659 | Gene W Yeo |
| National Institutes of Health | NS075449 | Gene W Yeo |

The funders had no role in study design, data collection and interpretation, or the decision to submit the work for publication.

## Author contributions

Sarah N Deffit, Conceptualization, Data curation, Formal analysis, Funding acquisition, Validation, Investigation, Methodology, Writing—original draft, Writing—review and editing; Brian A Yee, Emily C Wheeler, Data curation, Software, Formal analysis, Methodology, Writing—original draft, Writing—review and editing; Aidan C Manning, Data curation, Software, Formal analysis, Writing—review and editing; Suba Rajendren, Investigation, Methodology, Writing—review and editing; Pranathi Vadlamani, Michael C Washburn, Data curation, Writing—review and editing; Alain Domissy, Software; Gene W Yeo, Software, Supervision, Funding acquisition, Methodology, Writing—review and editing; Heather A Hundley, Conceptualization, Supervision, Funding acquisition, Methodology, Writing—original draft, Writing—review and editing

## Author ORCIDs

Heather A Hundley http://orcid.org/0000-0002-9106-9016

## Decision letter and Author response

Decision letter https://doi.org/10.7554/eLife.28625.016
Author response https://doi.org/10.7554/eLife.28625.017

# Additional files

## Supplementary files

• Supplementary file 1. A-to-I editing sites identified in neural cells. The high confidence editing sites identified by the bioinformatics pipeline are listed on the first sheet of the excel document (RNA-seq Identified Sites). The chromosome number (Column A) and coordinate in ce11 reference genome (Column B) are given for each predicted editing site. The approximate editing percentage (Column C) based on the frequency of reads with guanosine at that coordinate within unique reads as well as the number of unique reads covering that position (Column D) is listed. The predicted editing site was assigned (described in detail in the methods section) to a genic region (Column E) and a gene (Column F and G). A list of editing sites identified using Sanger sequencing editing assays from mRNAs identified by the bioinformatics pipeline are listed on the second sheet of the excel document (Sanger-seq Verification). Gene-specific reverse transcription followed by PCR amplification and Sanger sequencing was used to examine editing events in the indicated genes (Column A). The chromosome number (Column B) and coordinate in ce11 reference genome (Column C) are given for each adenosine to inosine detected as well as the percent editing as determined using the RNA-seq data (Column D). The methods used to detect the A-to-I change was listed (Column D and E) as well as confirmation (yes), decline (no), or inability to accurately determine (ND) the presence of A-to-I editing at a given adenosine. A list of all genes and the novel genes identified by the pipeline as edited are listed on the third sheet of the excel document (Edited Genes). All edited genes (Column A and B) were aligned with a document containing all identified editing sites in *C. elegans* from numerous published RNA-seq data sets (Supplementary file 3 [*Goldstein et al., 2017*]). Novel edited genes identified in this study are listed (Column C and D). Genes identified as edited by *SAILOR* were queried using Wormbase to identify genes that regulate chemotaxis and/or locomotion and these genes are listed on sheet four of the excel document (Locomotion and Chemotaxis Genes). The wormbase IDs and gene names (Column A and B) are listed for genes identified as regulators of this biological process. The genes were then aligned with an unpublished RNA-seq data set of RNAs bound by ADR-1 (Column C).

DOI: https://doi.org/10.7554/eLife.28625.013

• Supplementaty file 2. Differential gene expression identified from the transcriptome-wide RNA-seq. Genes whose transcripts exhibited ≥2 fold change in expression between wild-type and *adr-2(-)* neural cells are listed. Upregulated (Sheet 1) and downregulated (Sheet 2) genes are listed by gene name (Column A) and Wormbase ID (Column B). The base Mean, or mean expression of each gene normalized to sequencing depth for all samples is listed (Column C), as well as the fold change in expression observed when comparing the wild-type to *adr-2(-)* (Column D) and the adjusted p-value from DESeq2 (Column E). Genes whose expression was examined by qRT-PCR are marked with yellow and listed as verified (Column F). The four edited genes are listed as Edited (Column G).
DOI: https://doi.org/10.7554/eLife.28625.014

• Transparent reporting form
DOI: https://doi.org/10.7554/eLife.28625.015

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
