## [Decision Letter]

Thank you for submitting your article "The *C. elegans* neural editome reveals an ADAR target mRNA required for proper chemotaxis" for consideration by *eLife*. Your article has been reviewed by two peer reviewers, and the evaluation has been overseen by a Reviewing Editor and a Senior Editor. The following individual involved in review of your submission has agreed to reveal her identity: Brenda Bass (Reviewer #2).

The reviewers have discussed the reviews with one another and the Reviewing Editor has drafted this decision to help you prepare a revised submission.

Summary: While the genome-wide identification of ADAR RNA editing sites has been reported for multiple organisms and cell types, a tissue-specific analysis of *C. elegans* ADAR editing sites has not been performed. By adapting a previously described method, the authors isolate neural cells of L1-stage *C. elegans*, and using a novel pipeline they developed, define high confidence A-to-I editing sites in WT neural cells. The authors perform similar analyses on *adr-2* animals, which lack all editing, not only to validate editing sites, but also to determine genes that are differentially expressed. While *C. elegans* lacking ADARs were shown many years ago to have chemotaxis defects, the specific gene(s) responsible for this phenotype remains unknown. The authors go through a number of steps, both bioinformatic and experimental, to narrow their focus on differentially-expressed genes that are candidates for mediating the chemotaxis defects, and in the end, come up with a single candidate, the *clec-41* gene. The paper is well written, and the studies are well performed and nicely validated.

Essential revisions:

The reviewers and reviewing editor agree that a number of straightforward transgenic experiments are required to solidify the results of this study:

1) Rescue experiments (Figure 5) need a bit more detail. Expression of *clec-41* in a non-neuronal tissue would be a good control. The *clec-41* mutant should be included in the chemotaxis assays and rescue plasmids should be introduced into the *clec-41* mutant +/- the edited 3'UTR.

2) The rescue of the chemotaxis defect by expression of *clec-41* is an exciting result, and if true, would constitute a very significance advance to the field. However, to err on the side of caution, the authors should demonstrate the effect requires the open-reading frame of *clec-41*. Another possibility is that a different component of the *Prab-3::clec-41* array is responsible for the rescue of the *adr-2(-)* chemotaxis defects. For instance, RAB-3 has effects on chemotaxis, and an array carrying many copies of its promoter could influence *rab-3* expression and impact chemotaxis. Similarly, it is possible that the repetitive array produces dsRNA that sequesters RNAi factors. If the RNAi pathway is involved in regulating clec-41 expression as the authors suggest in the Discussion, sequestration of RNAi factors could also lead to rescue of chemotaxis. The authors used non-specific DNA to create their arrays, which should preclude this issue, but their transgene contains the *clec-41* 3' UTR, which presumably is double-stranded, and thus could also sequester RNAi factors. A straightforward experiment that would address these issues would be to replace the clec-41 open-reading frame with GFP, while keeping everything else in the array constant. If this negative control does not rescue chemotaxis, it would be definitive proof that CLEC-41 protein is a specific mediator of *adr-2* chemotaxis defects.

3) The mechanism by which defects in editing of *clec-41* 3'UTR cause reduced expression of *clec-41* are not explored at all. As it stands, the study is primarily an important gene discovery story (Figure 1–Figure 4) with one rescue experiment showing that *clec-41* over-expression rescues the chemotaxis defect in *adr-2* (Figure 5). Their data suggest that editing stabilizes *clec-41* expression in WT versus *adr-2*. A reporter-based system, where the 3'UTR of *clec-41* (+/- edited sites) is appended to GFP and driven more specifically in AWC neurons (e.g. *ceh-36* promoter) in WT and *adr-2* mutant would more directly address the question of 3'UTR involvement.

---

## [Author Response]

1) Rescue experiments (Figure 5) need a bit more detail. Expression of clec-41 in a non-neuronal tissue would be a good control. The clec-41 mutant should be included in the chemotaxis assays and rescue plasmids should be introduced into the clec-41 mutant +/- the edited 3'UTR.

The reviewers make a great point that expression of *clec-41* in a non-neural tissue would help determine if the CLEC-41 rescue of chemotaxis is cell autonomous or nonautonomous. To directly address this, we generated wild-type transgenic worms that express the *clec-41* transcript in the pharyngeal muscle using a *myo-2* promoter. The transgenic worms were then crossed to an *adr-2* mutant and progeny were screened to identify both wild-type and *adr-2(-)* worms expressing *myo-2* promoter driven *clec-41*. Chemotaxis assays were then performed and indicated that *myo-2* driven expression of *clec-41* was not capable of restoring chemotaxis to the *adr-2(-)* worms, suggesting neural specific expression of *clec-41* is required for proper chemotaxis (Figure 5—figure supplement 1).

We also agree that a good experiment would be to perform the rescue experiments in a *clec-41* mutant background. We obtained the only currently available *clec-41* mutant strain, *clec-41 (tm6722)*, from the Japanese National Bioresource Project; however, this strain was very slow growing and unable to be backcrossed. As backcrossing is necessary to eliminate any other mutations in the background, it is not possible to conduct experiments or interpret data generated using this mutant worm line.

2) The rescue of the chemotaxis defect by expression of clec-41 is an exciting result, and if true, would constitute a very significance advance to the field. However, to err on the side of caution, the authors should demonstrate the effect requires the open-reading frame of clec-41. Another possibility is that a different component of the Prab-3::clec-41 array is responsible for the rescue of the adr-2(-) chemotaxis defects. For instance, RAB-3 has effects on chemotaxis, and an array carrying many copies of its promoter could influence rab-3 expression and impact chemotaxis. Similarly, it is possible that the repetitive array produces dsRNA that sequesters RNAi factors. If the RNAi pathway is involved in regulating clec-41 expression as the authors suggest in the Discussion, sequestration of RNAi factors could also lead to rescue of chemotaxis. The authors used non-specific DNA to create their arrays, which should preclude this issue, but their transgene contains the clec-41 3' UTR, which presumably is double-stranded, and thus could also sequester RNAi factors. A straightforward experiment that would address these issues would be to replace the clec-41 open-reading frame with GFP, while keeping everything else in the array constant. If this negative control does not rescue chemotaxis, it would be definitive proof that CLEC-41 protein is a specific mediator of adr-2 chemotaxis defects.

We understand the reviewers’ concerns about whether the coding region or 3’ UTR present in the *rab3p::clec-41* array is mediating the rescue of the *adr-2* chemotaxis defects. To directly address this concern, we created a worm line where the *clec-41* open-reading frame was replaced with RFP and all other components are similar to the previous array (use of non-specific DNA and the same *rab3p::GFP::unc-54 3’ UTR* coinjection marker). The wild-type worms carrying this transgene were then crossed to *adr-2*(-) worms and the progeny were screened to identify both wild-type and *adr-2(-)* worms expressing *rab3p::RFP::clec-41 3’ UTR.* These worms were subjected to chemotaxis assays (Figure 5—figure supplement 1). The *adr-2(-)* worms expressing *rab3p::RFP::clec-41 3’ UTR* exhibited similar chemotaxis as the *adr-2(-)* worms, indicating that the *clec-41 3’ UTR* was not sufficient to rescue the *adr-2* chemotaxis defects.

3) The mechanism by which defects in editing of clec-41 3'UTR cause reduced expression of clec-41 are not explored at all. As it stands, the study is primarily an important gene discovery story (Figure 1–Figure 4) with one rescue experiment showing that clec-41 over-expression rescues the chemotaxis defect in adr-2 (Figure 5). Their data suggest that editing stabilizes clec-41 expression in WT versus adr-2. A reporter-based system, where the 3'UTR of clec-41 (+/- edited sites) is appended to GFP and driven more specifically in AWC neurons (e.g. ceh-36 promoter) in WT and adr-2 mutant would more directly address the question of 3'UTR involvement.

We thank the reviewers for suggesting to directly assess the effect of editing in regulating *clec-41* expression. To address this, we utilized a worm line with a genomic mutation (G184R) in the deaminase domain of the *adr-2* gene, which results in a glycine amino acid that is conserved across the ADAR family being converted to an arginine (Figure 6). We demonstrated that this worm line lacks A-to-I editing of *clec-41* (Figure 6). Importantly, this decreased editing is not due to a lack of ADR-2 protein binding the *clec-41* transcript in vivo as an ADR-2 RNA immunoprecipitation assay detected similar amounts of *clec-41* transcript in immunoprecipitates from wildtype worm lysates and those from the ADR-2 G184R mutant (Figure 6). To directly test whether the regulatory effect of ADR-2 on *clec-41* gene expression could be attributed to lack of RNA editing, we examined *clec-41* neural expression in the ADR-2 G184R mutant worm. Similar to the *adr-2(-)* worm line, *clec-41* expression in neural cells was significantly reduced in the ADR-2 G184R mutant line compared to wildtype worms, indicating that deamination is required for proper *clec-41* expression in neural cells (Figure 6). We then assessed chemotaxis in the ADR-2 G184R worms and observed a significant decrease in chemotaxis compared to wildtype worms, similar to that of the *adr-2(-)* worms (Figure 6). In sum, these data indicate that A-to-I editing is required for *clec-41* expression, which is important for proper chemotaxis.